# First Detection of Carbapenem-Resistant *Escherichia fergusonii* Strains Harbouring Beta-Lactamase Genes from Clinical Samples

**DOI:** 10.3390/pathogens8040164

**Published:** 2019-09-25

**Authors:** Tomilola Adesina, Obinna Nwinyi, Nandita De, Olayemi Akinnola, Emmanuel Omonigbehin

**Affiliations:** Department of Microbiology, College of Science and Technology, Covenant University, 112103 Ota, Nigeria; obinna.nwinyi@covenantuniversity.edu.ng (O.N.); manadina93@gmail.com (N.D.); ola.ayepola@covenantuniversity.edu.ng (O.A.); adedayo.omonigbehin@covenantuniversity.edu.ng (E.O.)

**Keywords:** *Escherichia fergusonii*, carbapenem resistance, extraintestinal pathogen, Extended-spectrum beta-lactamase (ESBL), multidrug resistance, *Escherichia coli*, Wound infections, beta-lactamase gene, antibiotics resistance

## Abstract

Recently discovered extraintestinal *Escherichia fergusonii* obtained from non-clinical samples has exhibited the potential for acquiring multiple beta-lactamase genes, just like many extraintestinal *Escherichia coli* strains. Albeit, they are often omitted or classified as *E. coli*. This study aimed to, therefore, identify carbapenem-resistant extended-spectrum beta-lactamase (ESBL) producing *E. fergusonii* isolates from clinical samples, determine their evolutionary relatedness using 16S rRNA sequencing analysis and screen for beta-lactamase genes. A total of 135 septic wound samples were obtained from patients on referral at a General Hospital in Lagos, Nigeria. For the phenotypic identification of isolates from culture-positive samples, morphological, and physiological tests were carried out. Identities of the isolates harbouring beta-lactamase genes were assigned to their genus strains using the 16S rRNA sequencing. The Kirby Bauer disc diffusion technique and double-disc synergy test were used to screen isolates for multidrug resistance and ESBL production. Carbapenem-resistant ESBL producing isolates were screened for beta-lactamase genes in a polymerase chain reaction. Three *E. fergusonii* isolates (CR11, CR35 and CR49) were obtained during this study. *E. fergusonii* strains were motile, non-lactose and non-sorbitol fermenting but positive for cellobiose and adonitol fermentation. The I6S rRNA assigned the phenotypically identified isolates to *E. fergusonii* species. All three isolates were multidrug-resistant, carbapenem-resistant and ESBL producers. Isolates CR11 and CR35 harboured cefotaximase (CTX-M) and temoniera (TEM) beta-lactamase genes while CR49 harboured sulfhydryl variable (SHV) beta-lactamase gene. We herein report the detection of multiple beta-lactamase genes in carbapenem-resistant ESBL producing *E. fergusonii* from clinical samples.

## 1. Introduction

Multidrug-resistant Extraintestinal pathogenic *E. coli* (ExPEC) clones have contributed significantly to the growing antibiotics resistance burden [1,2]. Several clinical multidrug-resistant ExPEC clones, including ST131, ST73 and ST40, reported in previous studies, have acquired resistance plasmids that encode multiple beta-lactamase genes, particularly those required for Extended-spectrum beta-lactamase (ESBL) production [3,4,5,6,7]. This facilitates their capacity in playing significant roles in the global dissemination of ESBL genes [1,8,9]. Extraintestinal *E. coli* ST131 clones (clade C1 and H30R) encoding CTX-M-15 genes have been reported in Japan, Thailand, Canada and Australia [10]. In Nigeria, several clones of multidrug-resistant ExPECs (ST131, ST295, ST617, ST501, ST448 and ST617) have also rapidly acquired transmissible plasmids that encode several beta-lactamase genes, including the CTX-M-15 genes [11].

Another extraintestinal human pathogen within the genus *Escherichia* has also shown tendencies for the dissemination of plasmid-mediated beta-lactamase genes. *Escherichia fergusonii* closely related to the ExPECs with about 60% similarity, was classified in 1985 as a new specie [7]. *Escherichia fergusonii* are opportunistic pathogens primarily associated with abdominal wounds, UTIs and bacteraemia in humans while causing septicaemia and diarrhoea in animals [7,12,13,14,15,16,17,18]. Not much is currently known about *E. fergusonii*. However, they are beginning to gain attention because several reports of multidrug-resistant *E. fergusonii* exists today. The first detection of ESBL production in clinical isolates of *E. fergusonii* was by Lagacé-Wiens et al. [19]. Although sulfhydryl variable (SHV-12) and CTX-M-1 conferring ESBL phenotypes have been reported in *E. fergusonii* from non-clinical samples, there is yet to be any report from human samples [7,20].

Most multidrug-resistant ExPEC isolated from bacterial infections in Nigeria are ESBL producers harbouring multiple beta-lactamase genes [21,22,23,24]. Moreover, recent studies on carbapenem-resistant Enterobacteriaceae infections in Nigeria have identified multidrug-resistant extraintestinal *Escherichia* sp. as a major causative organism [25,26,27]. However, identification of these organisms is often phenotypic based. We report herein, the 16S rDNA sequencing of carbapenem-resistant ESBL producing extraintestinal *Escherichia* spp. encoding multiple beta-lactamase genes.

## 2. Results

A total of three *E. fergusonii* isolates designated CR11, CR35 and CR49 were classified to distinguish them from other *Escherichia* species associated from wound infections. Isolates were motile, non-lactose fermenters, positive for cellobiose and adonitol but negative for sorbitol fermentation (Table 1). Following 16S rRNA sequencing analysis, the phenotypically identified isolates were assigned to *E. fergusonii* specie group. The isolates; CR11, CR35 and CR49 were renamed as *E. fergusonii* ADE4 (MH040118), *E. fergusonii* ADE11 (MH04009393) and *E. fergusonii* ADE18 (MH040100), respectively. In Figure 1, the phylogenetic tree constructed using partial 16S rDNA sequences of 1349 bp (*E. fergusonii* ADE4), 1390 bp (*E. fergusonii* ADE11) and 1385 bp (*E. fergusonii* ADE18), showed that all three isolates obtained from this study shared mostly the same close relatives. The sequence similarity percentages of *Escherichia fergusonii* ADE4 when compared with other 16S rRNA gene sequences in Genbank were 99.33% to *Escherichia fergusonii* ATCC 35469 (NR_027549.1), 99.33% to *Shigella flexneri* ATCC 29903, 99.26% to *Shigella sonnei* CECT 4887, 99.19% to *Escherichia fergusonii* NBRC 102419, 99.19% to *Escherichia fergusonii* ATCC 35469 (NR_074902) and 99.04% to *Escherichia* coli NBRC 102203. The 16S rRNA gene sequence similarity percentages of *Escherichia fergusonii* ADE11 were 99.56% to *Escherichia fergusonii* ATCC 35469 (NR_027549.1), 99.56% to *Shigella flexneri* ATCC 29903, 99.48% to *Escherichia fergusonii* ATCC 35469 (NR_074902.1), 99.48% to *Shigella sonnei CECT 4887*, 99.41% to *Escherichia fergusonii* NBRC 102419 and 99.33% to *Escherichia* coli NBRC 102203. For *Escherichia fergusonii* ADE18, sequence similarity percentages were 99.49% to *Escherichia fergusonii* ATCC 35469 (NR_027549.1), 99.49% to *Shigella flexneri* ATCC 29903, 99.42% to *Escherichia fergusonii* ATCC 35469 (NR_074902.1), 99.34% to *Escherichia fergusonii* NBRC 102419, 99.34% to *Escherichia* coli NBRC 102203 and 99.42% to *Shigella sonnei CECT 4887*.

Antibiotics susceptibility patterns of CR11, CR35 and CR49 based on CLSI interpretative criteria showed multidrug resistance. All three isolates exhibited 100% resistance to all beta-lactam antibiotics, including the extended-spectrum beta-lactam (ceftriaxone, cefuroxime and ceftazidime) and carbapenem (imipenem and meropenem) antibiotics. Isolates CR11 and CR35 were also resistant to other antibiotics class such as fluoroquinolones (levofloxacin, Ofloxacin, ciprofloxacin) and nitrofurantoin. All three isolates expressed ESBL production. Polymerase chain reaction (PCR) revealed that CR11, CR35 and CR49 encoded different beta-lactamase genes (Table 2). Isolates CR11 and CR35 harboured CTX-M type beta-lactamase (Figure 2) and TEM beta-lactamase genes while CR49 harboured SHV beta-lactamase gene. No carbapenemase gene was present in the three isolates from this study.

## 3. Discussion

This study to the best of our knowledge, reports the first isolation of carbapenem-resistant ESBL producing *E. fergusonii* of clinical importance, contrary to the first published study on the isolation of *E. fergusonii* from drinking water sources, in Nigeria [28]. Some studies suggest that *E. fergusonii* had existed long before the first identification in 1985 but were often omitted or misidentified as *E. coli*, due to the low discriminatory power of phenotypic methods [7,28]. Previous studies have also reported several inconsistencies with the use of biochemical methods including API 20E and Vitek 2 (Vitek System; bioMérieux, Marcy l’Etoile, France and bioMerieux, Inc., Durham, NC), for identification of *E. fergusonii* [29,30]. Recently, the ability to ferment cellobiose and adonitol and inability to ferment lactose and sorbitol among the *Escherichia* sp. has been reported to be specific for *E. fergusonii* and can be a guide for differentiating *E. fergusonii* from other members of the genus *Escherichia* [7]. *Escherichia fergusonii* isolates obtained from our study were motile, utilised cellobiose and adonitol but did not ferment lactose and sorbitol. The first published biochemical profile of *E. fergusonii* revealed that they are motile and non-lactose fermenters, confirming our findings [31]. The study of Balqis et al. [32] and Maheux et al. [16] however reported varying patterns for lactose fermentation and motility.

Not much information on phylogenetic studies of *E. fergusonii* are currently available, but phylogenetic analysis from this study revealed that *E fergusonii* ADE4, ADE11 and ADE18 shared the same clade with the genus *Escherichia* and *Shigella*. This is similar to previous reports on *Escherichia fergusonii* [31,33,34]. The genus *Escherichia* and *Shigella* are considered members of the same pathogenic lineage [35,36,37,38,39]. Naveena et al. [40] reported that *Shigella* and *E. coli* are phylogenetically the same but separated based on their clinical relevance and biochemical differences. *Shigella* and *Escherichia* are closely related genera with >99% 16S rDNA sequence similarity [40]. Whole-genome phylogeny analysis carried out by Sims and Kim [41], reported clustering of several members of the genus *Shigella* with different *Escherichia* phylogroups. Maheux et al., 2014 also reported the difficulty of using gene sequence alone in distinguishing non-*E. coli Escherichia* spp. strains in a group comprising *E. coli*, *Shigella* spp., *E. fergusonii*, and *E. albertii*. However, the use of a polyphasic identification method that combines 16S rDNA sequencing analysis with phenotypic methods has helped to differentiate *E. fergusonii* from *Shigella* sp., in this present study. Although both genera have very similar phenotypic traits, several differences in the biochemical characteristics of *E. fergusonii* have helped to differentiate them from *Shigella* sp. *Escherichia fergusonii* obtained from this study were motile, with the ability to ferment cellobiose and adonitol (Table 1), whereas members of the genus *Shigella* are non-motile, unable to ferment cellobiose and adonitol [42]. Also, *E. fergusonii* from this study were non-lactose fermenters (Table 1), unlike the slow lactose-fermenting *S. sonnei* [36].

The three carbapenem-resistant ESBL producing *E. fergusonii* stains obtained in this present study were resistant to imipenem, meropenem, ceftriaxone, ceftazidime and cefuroxime antibiotics. This finding is consistent with several studies that reported multiple resistance to broad-spectrum beta-lactam antibiotics in *E. fergusonii* isolated from animals and humans [17,19,31,43,44,45]. Although Savini et al. [17] reported that resistance to broad-spectrum cephalosporin antibiotics by non-ESBL producing *E. fergusonii* from their study was due to non-compliance of the patient to the use of cephalosporin antibiotics administered, the *E. fergusonii* strains from this study were ESBL producers. Hence, resistance to broad-spectrum cephalosporins observed in this study was likely due to the presence of beta-lactamase resistance genes detected in the *E. fergusonii* strains. *Escherichia fergusonii* ADE4 and ADE11 strains harboured CTX-M genes (See Table 2). Our finding is similar to the first report of *E. fergusonii* from a farm animal in South Korea, harbouring CTX-M gene [45]. Our study could, however, be the first documented report of CTX-M gene in *E. fergusonii* from human samples. The SHV and TEM beta-lactamase genes were also detected in *Escherichia fergusonii* ADE18 and *Escherichia fergusonii* ADE4/ADE11, respectively. This corresponds with the study that reported SHV-12 as the first beta-lactamase gene detected in *E. fergusonii* from a human specimen [19]. Resistance to imipenem and meropenem antibiotics was observed as well in this study, supporting the first report on carbapenem-resistant *E. fergusonii* strains among 23 *E. fergusonii* isolates obtained from non-human primates [7]. The first report of carbapenem-resistant *E. fergusonii* from human samples is likely to be this study. None of the screened carbapenemase genes was present within the three *E. fergusonii* strains from this study. Resistance to carbapenem antibiotics could have been influenced by the beta-lactamase genes harboured by the *E. fergusonii* strains or was conferred by other carbapenemase gene types (e.g. OXA-48) not investigated in this study.

The limitation of this study was due to limited resources. Beta-lactamase genes detected during this study were not sequenced to determine their spectrum of activity and some carbapenemase genes were not screened. Despite these limitations, this study has shown that *E. fergusonii* can be a reservoir for the spread of antibiotics resistance and a threat to the effective treatment of bacterial infections they cause.

## 4. Materials and Methods 

### 4.1. Culture and Identification of Clinical Isolates

#### 4.1.1. Culture and Phenotypic Identification of Clinical Isolates

A total of 135 samples were obtained from septic wound patients on referral to Government general hospital Odan, Lagos, Nigeria. Wound swabs were obtained using sterile swab sticks, labelled appropriately and transported to the laboratory within 2-4 h of collection for further analysis. Samples were cultured on sterile MacConkey agar plates (Rapid Lab, UK), using the streak plate method and incubated for 24 h at an optimum temperature of 37 °C. Positive cultures were subcultured on MacConkey agar to obtain pure isolates [46]. All isolates were phenotypically identified following standard routine morphological and cultural characteristics, as well as biochemical tests such as Gram stain reaction, sulphur reduction, indole, motility, methyl red, voges-Proskauer, urease, oxidase, sugar fermentation and citrate utilisation test [46]. Pure isolates phenotypically identified as *E. coli*. were stored in 20% Luria-Bertani (LB) broth (HIMEDIA, India) at −80 °C for long-term storage while working stocks for further analysis were held at 4 °C. *Escherichia* sp. were further screened for the ability to utilise cellobiose, sorbitol and adonitol. Motile isolates positive for cellobiose and adonitol fermentation but negative for lactose and sorbitol utilisation were designated CR11, CR35 and CR49.

#### 4.1.2. Genotypic Identification of Isolates

A discrete colony of CR11, CR35 and CR 49 in overnight broth cultures were used for the 16S rRNA gene sequencing method [47]. Genomic DNA of isolates CR11, CR35 and CR 49 were extracted using a commercial genomic DNA extraction kit (AidLab, China), following manufacturer’s instruction. Bacteria universal primer (27F: f5′-AGAGTTTGATCCTGGCTCAG-3′ and 1492R: r5′-GGTTACCCTTGTTACGACTT-3′) were used to amplify 16S rRNA gene of selected carbapenem-resistant ESBL producing isolates, in a simplex polymerase chain reaction (PCR), as described previously [48]. Briefly, initial denaturation of PCR mixture was carried out at 94 °C for 5 min, followed by 35 cycles at 94 °C for 30 s, 52 °C for 30 s, 68 °C for 60 s and a final extension at 68 °C for 5 min. 30 s. The PCR products were analysed by electrophoresis in Tris-borate EDTA buffer and viewed using UV fluorescence in a gel image documentation system. PCR pleural was purified using a PCR clean-up kit (Zymo Research, U.S.A.) [48] and eluted for sequencing at Macrogen Corp, Maryland, USA. Nucleotide sequences were determined using automated sequencer (Applied Biosystems SeqStudio Genetic Analyzer) and the dye-deoxy termination procedure. All 16S rRNA gene sequences obtained were edited using Bioedit (version 7.0.26) software and aligned by multiple sequence alignment technique using CLUSTAL W. The obtained bacterial DNA sequence was compared with other 16S rRNA genes in the GenBank, using the NCBI Basic Local alignment search tools BLAST-n program and a phylogenetic tree was constructed using Neighbour-Joining method in MEGA version 7.0.26 according to Saitou and Nei [49].

### 4.2. Antimicrobial Susceptibility Testing

Antimicrobial susceptibility patterns of CR11, CR35 and CR49 were determined by the disc diffusion method according to the Clinical and Laboratory Standards [50]. Multiple discs (Rapid Labs, UK) containing Ceftazidime (CAZ, 30 µg), Cefuroxime (CXM, 30 µg), Cefotaxime (CTX, 30 µg), Nitrofurantoin (ATM, 300 µg), Gentamicin (GEN, 10 µg), Ciprofloxacin (CIP, 5 µg), Ofloxacin (OFX, 5 µg), Augmentin (AUG, 30 µg) and Ampicillin (10 µg) antibiotics, frequently used within the community and healthcare setting in Nigeria, were tested. Single discs of Ceftriaxone (CRO, 30 µg), penicillin G (PEN G, 10 µg), imipenem (10 µg), Ertapenem (10 µg) and meropenem (10 µg) purchased from Oxoid^TM^ (United Kingdom) were also assessed. In brief, isolate suspensions diluted using 0.5 McFarland standard were streaked with a sterile cotton swab on freshly prepared Mueller Hinton agar (MHA) plates. Multiple and single antibiotics discs (Rapid Labs, UK) to be tested were carefully placed on the seeded plate with the aid of a sterile forceps. Plates were inverted and transfer into the incubator for 15–18 h at 37 °C. *Escherichia coli* ATCC 25922 was used as control strains following CLSI recommendation [50]. Results of antimicrobial susceptibility testing were interpreted as sensitive, intermediate or resistant, based on the zone diameter interpretive standard for Enterobacteriaceae, as published by the Clinical Laboratory Standards Institute [50].

### 4.3. Detection of ESBL Production

Double disc synergy test (DDST) was carried out using Oxoid^TM^ (United Kingdom) purchased augmentin (amoxicillin-clavulanate) and extended-spectrum cephalosporin antibiotics, following the recommended European Committee on Antimicrobial Susceptibility Testing breakpoint [51,52]. Ceftazidime (CAZ, 10 µg) and cefotaxime (CTX, 5 µg) disc alone and in combination with amoxicillin-clavulanate (AMX, 20/10 µg) were used to detect ESBL activity in isolates with reduced susceptibility to extended-spectrum cephalosporins. Mueller-Hinton agar (MHA) plates prepared according to manufacturer’s standard were seeded with isolate (CR11, CR35 and CR49) suspensions of 0.5 McFarland turbidity standard. Antibiotic discs were placed with sterile forceps 25mm apart, centre to centre of the plates and incubated aerobically for 18 h at 37 °C. *Escherichia coli* ATCC 25922 strain was used as control. ESBL results were interpreted following the zone diameter interpretive clinical standard of the European Committee on Antimicrobial Susceptibility Testing. An increased inhibition zone diameter of ≤ 20mm centre to centre, augmented in the direction of amoxicillin-clavulanate antibiotics was recorded positive for ESBL production.

### 4.4. DNA Extraction and Genotypic Detection of β-lactamase Genes

DNA was extracted from CR11, CR35 and CR 49. DNA extraction was carried out using TENS (Tris-HCl; 10 mM (pH 8.0), EDTA; 1 mM, NaOH; 0.1 N, SDS; 0.5% (w/v)) miniprep [53,54]. Discrete colonies of overnight cultures were inoculated into 1.5 mL of Luria Bertani (LB) broth (HIMEDIA, India) in 2 mL Eppendorf tubes and incubated overnight for 18 h at 37 °C. Eppendorf tubes containing overnight cultures were spinned in a C1008-C centrifuge (Benchmark Scientific, USA) at 10,000 rpm for 1 min, to harvest cells. Supernatants were discarded, and bacterial pellets were resuspended in 1000 μL of phosphate-buffered saline (PBS), to remove cellular debris. Tubes containing resuspended cells were centrifuged at 10,000 rpm for 1 min, and supernatants were discarded. Aliquots of 300 μL of TENS buffer were added to bacterial pellets, gently inverted 3-4 times, to lyse bacterial cells. Aliquots of 150 μL 3M NaOAc (pH 5.6) were added, and the tubes were gently inverted 3-4 times to precipitate DNA. The tubes were centrifuged at 13,000 rpm for 5 min to pellet white precipitates. Clear supernatants containing DNA were carefully removed and transferred into new sterile Eppendorf tubes, using sterile 200 μL pipettes. Aliquots of 900 μL of 95% ethanol were added to clear supernatants and tubes were inverted. The tubes were centrifuged at maximum speed for 2 min to pellet DNA. Supernatants were discarded, and 500 μL of 70% ethanol were pipetted into tubes to wash the DNA pellets. The obtained DNA pellets were centrifuged for 2 min, ethanol solutions were discarded, and DNA pellets were dried in a CentriVap DNA [53,54] concentrator 79820 (Labconco, USA). DNA was dissolved in 50 μL 10mM Tris-HCl (pH 8) and chilled on ice for further experiment. DNA was screened for eight different beta-lactamase genes, including ESBLs (SHV, TEM and CTX-M genes), Oxacillinase (OXA) and carbapenemases (IMP, VIM, KPC and NDM genes). Beta-lactamase genes were screened for using specific primers (Table 3) as previously described [55,56,57,58]. For polymerase chain reaction (PCR) amplification of beta-lactamase genes in a thermocycler C100 Touch (Bio-RAD, USA), 5 μL aliquot of DNA template was added to a 20 μL master mix. Master mix contained 0.5 μL of deoxynucleotide (dNTPs) solution (New England Biolabs, UK), 0.5 μL each of primer pairs (forward and reverse), 0.125 μL *Thermus aquaticus* YT-1 (Taq) DNA polymerase (New England Biolabs, UK), 2.5 μL 10x standard *Taq* reaction buffer (New England Biolabs, UK) and 16.375 μL nuclease-free water (New England Biolabs, UK). The PCR mixture was preheated at 94 °C for 5 min, followed by 35 cycles at 94 °C for 30 s, 30 s at annealing temperature (Table 3), 60 s at 72 °C, and 72 °C for 7 min. About 4 μL of amplified PCR mixture was resolved on agarose (1%) gel electrophoresis stained with 4 μL ethidium bromide. Expected amplicon size was visualised under a UV fluorescence in a gel image documentation system 220 (Bio-Rad, UK).

### 4.5. Statistical Analysis

Results from this study were analysed using the SPSS version 20.0 statistical package. Results were presented in tables, figures and dendrogram.

## 5. Conclusions

The results of 16S rRNA gene sequencing analysis and phylogeny were consistent with the phenotypic identification of carbapenem-resistant ESBL-producing strains obtained from this study. However, for molecular identification and determination of evolutionary relationships, the 16S rRNA gene analysis method alone was not sufficient to adequately distinguish *E. fergusonii* stains from other closely related species. The use of partial 16S rDNA sequences (1349–1390 bp) of the three isolates for the construction of the phylogenetic tree might have contributed to the difficulty associated with differentiating *Shigella* from *E. fergusonii*, genotypically in this study. Further molecular analysis such as DNA-DNA hybridisation or multilocus sequence typing (MLST) is recommended to complement 16S rRNA gene analysis, for revealing molecular differences at the species level. Molecular characterisation of resistance genes detected during this study will as well throws more light on their location and clinical relevance.

## Figures and Tables

**Figure 1 pathogens-08-00164-f001:**
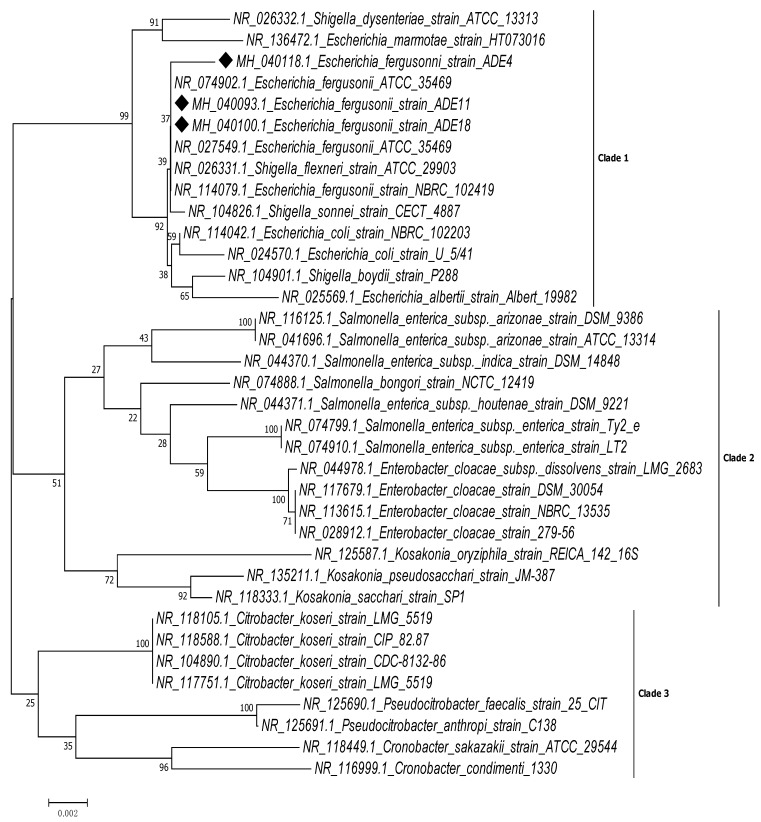
Phylogenetic tree showing the relationship of *E. fergusonii* strains ADE4, ADE11 and ADE18 from wound infection with other closely related members of Enterobacteriaceae, based on the similarities in their 16S rRNA gene sequences. The phylogenetic tree was reconstructed by comparing 36 concatenated sequences using the neighbour-joining (NJ) method while evolutionary distances were computed with the Tamura 3-parameter model, in MEGA ver. 7.0.26 software.

**Figure 2 pathogens-08-00164-f002:**
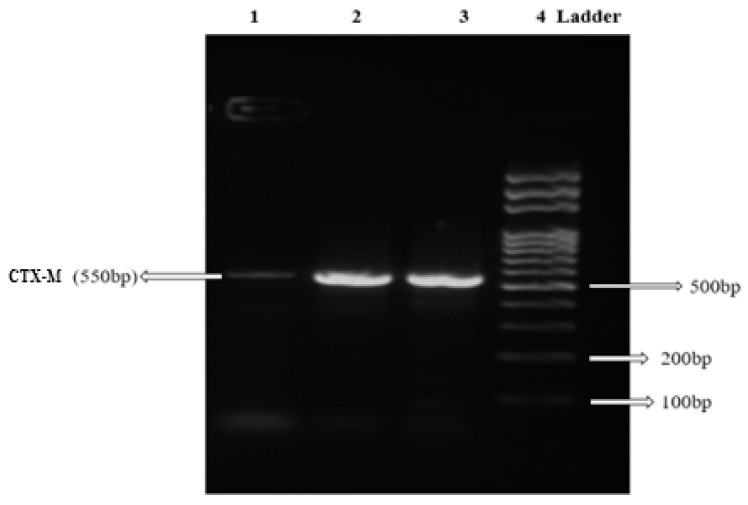
Agarose gel electrophoresis plate showing CTX-M extended-spectrum beta-lactamase (ESBL) genes under an ultraviolet (UV) transilluminator: Lane 1- Positive control (*K. pneumoniae* ATCC 7881); Lane 2- CTX-M from CR11; Lane 3- CTX-M from CR35; and Lane 4-Molecular weight marker (Norgen, 100 bp DNA Ladder, Canada).

**Table 1 pathogens-08-00164-t001:** Morphological and biochemical characteristics of isolates.

Organism Name		Form		Margin	Elevation	Optical Property		Appearance		Consistency		Colour on NA	Colour on MacC		Gram Stain
***E. fergusonii***		circular		Entire	Flat	opaque		Glistening		Moist		White	Pink		Negative
**Indole production**	**Methyl-red**	**Voges-Proskauer**	**Citrate**	**Urease**	**Hydrogen sulphide**	**Motility**	**Lactose**	**Maltose**	**Sucrose**	**Glucose (A/G)**	**Mannitol**	**Cellobiose**	**Adonitol**	**Sorbitol**	**Oxidase**
**+**	**+**	**−**	**−**	**−**	**−**	**+**	**−**	**+**	**−**	**+/+**	**+**	**+**	**+**	**−**	**−**

Keys: NA; Nutrient agar Mac; MacConkey agar A; Acid G; Gas +; positive −: negative.

**Table 2 pathogens-08-00164-t002:** Antimicrobial susceptibility patterns and associated beta-lactamase genes of isolates CR11, CR35 and CR49.

Isolate Name	Organism	Accession Number	Antibiotics Resistance Pattern	Beta-Lactamase Gene Encoded
CR11	*E. fergusonii* **ADE4**	MH040118	AUG PEN G AMP CRO CRX CAZ LEV OFL CIP NIT GEN IMP MER	TEM CTX-M
CR35	*E. fergusonii* **ADE11**	MH040093	AUG PEN G AMP CRO CRX CAZ LEV OFL CIP NIT GEN IMP MER	TEM CTX-M
CR49	*E. fergusonii* **ADE18**	MH040100	AUG PEN G AMP CRO CRX CAZ IMP	SHV

Key: AMP: ampicillin, PEN G; penicillin G, AUG: augmentin, CRO: cefuroxime, CRX: ceftriaxone, CAZ: ceftazidime, CTX: cefotaxime, LEV: levofloxacin, OFL: ofloxacin, GEN: gentamicin, NIT: nitrofurantoin, CIP: ciprofloxacin.

**Table 3 pathogens-08-00164-t003:** Oligonucleotide primers used for beta-lactamase gene detection.

Primer Name	Primer Sequence 5′ ⟶ 3′	Annealing Temperature	Expected Amplicon Size	Reference
TEM	F:5′CGCCGCATACACTATTCTCAGAATGA3′ R:5′ACGCTCACCGGCTCCAGATTTAT3′	58 °C	445 pb	[55]
SHV	F:5′CTTTATCGGCCCTCACTCAA3′ R:5′AGGTGCTCATCATGGGAAAG3′	58 °C	273 bp	[55]
CTX-M	F:5′CGCTTTGCGATGTGCAG3′ R:5′ACCGCGATATCGTTGGT3′	55 °C	550 bp	[56]
IMP	F:5′-GGAATAGAGTGGCTTAAYTCTC-3′ R:5′-CCAAACYACTASGTTATCT-3′	53 °C	188 bp	[59]
NDM	F:5′GGTTTGGCGATCTGGTTTTC R:5′CGGAATGGCTCATCACGATC	62.8 °C	621 bp	[57]
OXA	F:5′ACACAATACATATCAACTTCGC3′ R:5′AGTGTGTTTAGAATGGTGATC3′	62.8 °C	813 bp	[55]
KPC	F:5′GTATCGCCGTCTAGTTCTGC3′ R:5′GGTCGTGTTTCCCTTTAGCCA3′	59.9 °C	636 bp	[60]
VIM	F:5′-GATGGYGTTTGGTCGCATATCKCAAC3′ R:5′-CGAATGCGCAGCACCRGGATAGAA-3′	54.4 °C	390 bp	[57]

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
