# Peer review of "First Detection of Carbapenem-Resistant Escherichia fergusonii Strains Harbouring Beta-Lactamase Genes from Clinical Samples"

_pathogens, 2019, doi:10.3390/pathogens8040164_

Round 1
Reviewer 1 Report
In general the manuscript is a good. The topics is really hot.
I recomand a few modification.
Double check the english.
Row 48,64, 90 - please describe the abbreviations when you use them for the first time
Row 134 please reformulate the table no 2 name... positive culture not culture positive
row 185 -please finish your sentence, or reformulate.
row 198-the correct test name is Vitek 2, not Viteck 2.You also have to give details about the country and the name of the company that produces the tests
Thanks for the oportunity of reading the article
Author Response
Response to Reviewer 1:
We sincerely appreciate the effort of the reviewer for their detailed review and constructive comments. Indeed, all the comments have culminated in improving the quality of this manuscript. Our responses to all comments are as follows:
Comment 1: Double check the English
Reply: Manuscript has been edited for errors with use of English and addressed
appropriately.
Comment 2: Row 48,64, 90 - please describe the abbreviations when you use them for the first time.
Reply: The suggested correction has been made.
Comment 3: Row 134 please reformulate the table no 2 name... positive culture not culture positive
Reply: The correction has been made.
Comment 4: Row 185 -please finish your sentence, or reformulate.
Reply: The correction has been made.
Comment 5: Row 198-the correct test name is Vitek 2, not Viteck 2. You also have to give details about the country and the name of the company that produces the tests
Reply: The suggested correction has been made.
Thank you.

Reviewer 2 Report
Comments to author
This manuscript isolated and characterized the carbapenem-resistant E. fergusonii strains showing multiple beta lactamase genes from clinical samples. Overall, I find this manuscript interesting and well written but very descriptive. However, in my opinion, some of the results shown in the manuscripts need further control experiments or not well represented to have a better conclusion. I have some suggestions with respect to the presentation, incorporation of which will help the reader to understand in a better way and without these controls experiment it is hard to conclude the observation made by the authors.
My comments are listed below.
1. The figure 1 should be run with a positive control with CTX-M15 gene. The legend in the figure 1 is misleading. What does lane #3-5 denotes? what the negative control denotes should be explained properly. There in no lane #20 but in the figure legend Lane 20- Marker. I would suggest to re-run the gel and replace the figure with only those lanes which is descriptive, and a positive control is a must to make the authors’ point for the figure.
2. Line 131-133: the author has mentioned the drug resistance phenotype to imipenem and meropenem. Which figure they are referring? There is no data shown for this observation in any of the tables. Same thing is talked in the line number 217-219. If there is data, then always refer to that part for the convenience of reader.
3. Again, in line number 195-196 without showing the data about motility and fermenting lactose authors have made a statement. I would recommend including the data as a figure.
4. The discussion section is too long and written more like a review rather than explaining the observations. I would recommend cutting it to make an impact on the reader.

Author Response
Response to Reviewer 2:
We sincerely appreciate the effort of the reviewer for their detailed review and constructive comments. Indeed, all the comments have culminated in improving the quality of this manuscript. Our responses to all comments are as follows:
Comment 1:
The figure 1 should be run with a positive control with CTX-M15 gene. The legend in the figure 1 is misleading. What does lane #3-5 denote? what the negative control denotes should be explained properly. There is no lane #20 but in the figure legend Lane 20- Marker. I would suggest to re-run the gel and replace the figure with only those lanes which is descriptive, and a positive control is a must to make the authors’ point for the figure.
Reply: A new agarose gel picture has been used to replace the previous one.
Comment 2:
Line 131-133: the author has mentioned the drug resistance phenotype to imipenem and meropenem. Which figure they are referring? There is no data shown for this observation in any of the tables. Same thing is talked in the line number 217-219. If there is data, then always refer to that part for the convenience of reader.
Reply: A new table showing carbapenem resistance pattern has been included in the manuscript.
Comment 3:
Again, in line number 195-196 without showing the data about motility and fermenting lactose authors have made a statement. I would recommend including the data as a figure.
Reply: A new table showing morphological and biochemical characteristics of
Escherichia sp. obtained during this study has been included in the
manuscript.
Comment 4:
The discussion section is too long and written more like a review rather than explaining the observations. I would recommend cutting it to make an impact on the reader.
Reply: The discussion section has been edited and reduced appropriately.
Thank you.

Reviewer 3 Report
The authors provide information on multidrug resistant and presumptive E. fergusonii strains identified from clinical samples in Nigeria, where ESBL producing Enterobacteriaceae (including E. coli) are very common. Their perspective is to provide epidemiological and molecular data on antibiotic resistance profiles of isolates from a species for which the knowledge is still limited in terms of ecology, phenotypic and genotypic properties. However, there are several serious constraints that raise questions on the validity of the results and the conclusions presented. Main concerns are:
- Species identification. 16S RNA is not a reliable marker to differentiate bacteria at the species level (including Escherichia sp.) since many species share 98-99% nucleotide similarity (see for example doi: 10.1099/ijs.0.000228.
- Beta-lactamase content. TEM or SHV group of beta-lactamases comprise a wide diversity of enzymes with variable hydrolytic profiles. Many of them are not extended-spectrum beta-lactamases and some (such as TEM-1) are intrinsic to some species. This is especially important in strain CR49 (where only SHV was detected). It is critical to confirm the relationship between the phenotype and the genotype since extended-spectrum beta-lactam resistance might be conferred by non-enzymatic mechanisms. Also, how are the authors certain of the presence of CTX-M-15 if the product of PCR was only 550bp long? The blaCTX-M-1 group genes are distinguished by mutations all over the gene.
- Antibiotic resistance profiles. Also linked to the previous comment, it is expected that E. fergusonii strains are resistant to penicillin (see first species description JCM 1985), so for the purpose of the study, the authors need to clearly focus on acquired resistance profiles.
- Carbapenem resistance. There is a large misunderstood here. At some points in the text, the authors refer that ESBL are determinants of carbapenems resistance (e.g. beginning of third paragraph of discussion). This is definitely not true. In the absence of clarification on molecular mechanisms for this phenotype, it is possible that the combination of ESBL production plus porins’ defects might be contributing to this phenotype, but the authors need to reformulate all this part of the text.
- Discussion section. Several additional assumptions are made in the discussion that are not adequately supported by the results presented (e.g. plasmid location of beta-lactamase genes, similarity with previous findings).
Minor points:
Abstract. The first and last sentence are somewhat contradictory.
The text is excessively long in some parts and could be significantly focused and reduces (e.g. introduction on E. coli and discussion on Shigella could be shortened).
Table 5. “Accession number” instead of “Ascension number”.
Author Response
Response to Reviewer 2:
We sincerely appreciate the effort of the reviewer for their detailed review and constructive comments. Indeed, all the comments have culminated in improving the quality of this manuscript. Our responses to all comments are as follows:
Comment 1:
Species identification. 16S RNA is not a reliable marker to differentiate bacteria at the species level (including Escherichia sp.) since many species share 98-99% nucleotide similarity (see for example doi: 10.1099/ijs.0.000228.
Reply: We acknowledge that 16S rRNA sequencing alone is not conclusive in determining the evolutionary relatedness of the organism. However, we combined phenotypic and molecular characterization methods for the identification of the organisms. The technique (combined phenotypic and molecular methods) is still reliable because it actually reveals the direct functional properties of the isolated organisms and its evolutionary relatedness. This technique is still being used even at the Centers for Disease Control and Prevention (CDC) reference laboratory up till today. Also, recent studies have reported the ability of some Escherichia sp. to breakdown certain sugars such as adonitol and cellobiose, to be specific to E. fergusonii. Fermentation of these sugars has been included in this study for proper identification.
Comment 2:
Beta-lactamase content. TEM or SHV group of beta-lactamases comprise a wide diversity of enzymes with variable hydrolytic profiles. Many of them are not extended-spectrum beta-lactamases and some (such as TEM-1) are intrinsic to some species. This is especially important in strain CR49 (where only SHV was detected). It is critical to confirm the relationship between the phenotype and genotype since extended-spectrum beta-lactam resistance might be conferred by non-enzymatic mechanisms. Also, how are the authors certain of the presence of CTX-M-15 if the product of PCR was only 550bp long? The blaCTX-M-1 group genes are distinguished by mutations all over the gene.
- Antibiotic resistance profiles. Also linked to the previous comment, it is expected that E. fergusonii strains are resistant to penicillin (see first species description JCM 1985), so for the purpose of the study, the authors need to clearly focus on acquired resistance profiles.
Reply: All resistant gene detected during this study were encoded on plasmids. From literature studies, most plasmid-mediated TEM and SHV type beta-lactamases are variants of the parent type (TEM1 and SHV1/SHV2) and usually code for ESBL production. However, since sequencing analysis helps to differentiate TEM and SHV variants, we have proposed further studies such as the use of sequencing analysis. Again, the primer used during this study was specific for CTX-M-15 since only sequencing can ascertain CTX-M variants, we have changed CTX-M-15 to CTX-M until further studies.
Comment 3:
Carbapenem resistance. There is a large misunderstood here. At some points in the text, the authors refer that ESBL are determinants of carbapenems resistance (e.g. beginning of third paragraph of discussion). This is definitely not true. In the absence of clarification on molecular mechanisms for this phenotype, it is possible that the combination of ESBL production plus porins’ defects might be contributing to this phenotype, but the authors need to reformulate all this part of the text.
Reply:
We have rephrased the statement above to communicate our point appropriately.
Comment 4:
Discussion section. Several additional assumptions are made in the discussion that are not adequately supported by the results presented (e.g. plasmid location of beta-lactamase genes, similarity with previous findings).
Reply: This section has been edited and discussed appropriately.
Comment 5:
Minor points:
Abstract. The first and last sentence are somewhat contradictory.
The text is excessively long in some parts and could be significantly focused and reduces (e.g. introduction on E. coli and discussion on Shigella could be shortened).
Table 5. “Accession number” instead of “Ascension number”.
Reply: All concerns have been addressed appropriately.
Thank you.
Reviewer 4 Report
This is a well-written paper with minor grammar errors. Quality of presentation was excellent with remarkable scientific soundness.
A few suggestions:
Could strains be compared exclusively to other 16s published genomes (draft or whole) of E. fergusonii to determine level of relatedness? eg: GTA-EF02 - GTA-EF04 sequences on NCBI? It would be of scientific interest to know how they group rather than limiting it to the 11 used in this comparative analysis. Or perhaps an explanation on the choice of the 11?
"Escherichia fergusonii ATCC 35469 and NBRC 102419 were the closest relatives to E fergusonii ADE4 (95.38%), ADE11 (98.06%) and ADE18 (97.55%). Other close relatives of ADE4 (95%), ADE11 (98%) and ADE18 (95%) were Escherichia vulneris, Shigella dysenteriae, Escherichia. coli, Shigella flexneri and Shigella sonnei."
I'm curious about the %relatedness of the closest relative of ADE11, as from the tree, Shigella was closest. Perhaps this text should be re-phrased?
These are minor concerns but would add great value to the paper.
Author Response
Response to Reviewer 4:
We sincerely appreciate the effort of the reviewer for their detailed review and constructive comments. Indeed, all the comments have culminated in improving the quality of this manuscript. Our response to the comment is as follows:
Comment: Could strains be compared exclusively to other 16s published genomes (draft or whole) of E. fergusonii to determine the level of relatedness? eg: GTA-EF02 - GTA-EF04 sequences on NCBI? It would be of scientific interest to know how they group rather than limiting it to the 11 used in this comparative analysis. Or perhaps an explanation on the choice of the 11?
Reply: All three isolates have been compared exclusively with other similar sequences in Genbank. More sequences have been included to construct a more robust phylogenetic tree.
Thank you for the suggestion. The phylogenetic tree looks more meaningful.
Round 2
Reviewer 2 Report
Response to Reviewer 2:
We sincerely appreciate the effort of the reviewer for their detailed review and constructive comments. Indeed, all the comments have culminated in improving the quality of this manuscript. Our responses to all comments are as follows:
Comment 1:
The figure 1 should be run with a positive control with CTX-M15 gene. The legend in the figure 1 is misleading. What does lane #3-5 denote? what the negative control denotes should be explained properly. There is no lane #20 but in the figure legend Lane 20- Marker. I would suggest to re-run the gel and replace the figure with only those lanes which is descriptive, and a positive control is a must to make the authors’ point for the figure.
Reply: A new agarose gel picture has been used to replace the previous one.
Reviewer: The author state that a new gel picture is included but I see it’s the same gel image with just lanes labeled in the figure legend. It would be better for the readers that author run a new agarose gel with just Lane 1, lane 2 and lane 7 and include a positive control in which they can take a template from a bacterium which has CTX-M15 gene. Also, if they want to show the no band lanes on the gel then in figure legend they should mention which strain was used for template and they see no CTX-M15 gene. What is used in lane 3-5 for running PCR is not clear from the figure still. Please replace the agarose gel by running a new one not just edit the same gel.
Comment 2:
Line 131-133: the author has mentioned the drug resistance phenotype to imipenem and meropenem. Which figure they are referring? There is no data shown for this observation in any of the tables. Same thing is talked in the line number 217-219. If there is data, then always refer to that part for the convenience of reader.
Reply: A new table showing carbapenem resistance pattern has been included in the
manuscript.
Reviewer: Satisfied
Comment 3:
Again, in line number 195-196 without showing the data about motility and fermenting lactose authors have made a statement. I would recommend including the data as a figure.
Reply: A new table showing morphological and biochemical characteristics of
Escherichia sp. obtained during this study has been included in the
manuscript.
Reviewer: How the authors have determined the strains are motile or non-motile? Show the data rather than just putting a table for it. A simple way to show the motility is the use of low agar plate and checking the growth pattern after overnight incubation. The plate image date should be included along with the table.
Comment 4:
The discussion section is too long and written more like a review rather than explaining the observations. I would recommend cutting it to make an impact on the reader.
Reply: The discussion section has been edited and reduced appropriately.
Thank you.
Reviewer: Satisfied

Author Response
Round 2 Response
Comment 1:
Reviewer: The author state that a new gel picture is included but I see it’s the same gel image with just lanes labeled in the figure legend. It would be better for the readers that author run a new agarose gel with just Lane 1, lane 2 and lane 7 and include a positive control in which they can take a template from a bacterium which has CTX-M15 gene. Also, if they want to show the no band lanes on the gel then in figure legend they should mention which strain was used for template and they see no CTX-M15 gene. What is used in lane 3-5 for running PCR is not clear from the figure still. Please replace the agarose gel by running a new one not just edit the same gel.
Reply: Previous gel has been replaced by running a new gel.
Comment 2:
Reviewer: How the authors have determined the strains are motile or non-motile? Show the data rather than just putting a table for it. A simple way to show the motility is use of low agar plate and checking the growth pattern after overnight incubation. The plate image date should be included along with the table.
Reply: Motility test was determined using SIM (Sulphur Indole Motility) agar.
Thank you.

Reviewer 3 Report
I recognize the effort of the authors to further explain or solve most problems raised in the previous review, though this is not consistent along the text. However, there are still some concerns that need to be corrected before publication:
1. Strains identification. The 16S data provided to support identification is not reliable. First, the fragment analyzed is only ~700bp long which is insufficient to provide appropriate discriminatory power. Second, the comparison of 16S sequences obtained and deposited in GenBank reveals a ~90% identity between the three isolates (with several gaps or NT basis that need to be revised and corrected) (see clustal W attached). Third, according to the authors, the 16S sequences obtained showed identical homologies (e.g. ADE4 95%) with either the type strain from E. fergusonii or Shigella sp. or other Escherichia species. So, the authors are advised to provide a larger 16S sequence (~1.5Kb) from each isolate that would provide a higher resolution and support phenotypic data. These sequences should substitute those already available on GenBank.
2. ESBL characterization. PCR performed on plasmid DNA extracted are not sufficient to support the presence of plasmid-mediated beta-lactamases. The extracts can have chromosomal DNA (that in most cases is not completely eliminated) which is a confounding factor. Again, other narrow spectrum TEM or SHV variants might be present. As such, sequencing of these products needs to be performed for this study and not in subsequent studies because it is needed to support results and conclusions. Otherwise, the authors need to recognize this limitation and modify the manuscript in accordance.
3. Carbapenems resistance. OXA-48 carbapenemase genes were not searched by PCR. The primers used to search for OXA are directed for OXA-1 variants that include either narrow and extended-spetrum beta-lactamases. If the authors decide not to search for OXA-48 type carbapenemases, they need to consider this limitation in the manuscript.
Other comments:
Abstract. The identity of the strains should be considered immediately before presenting their phenotypic properties.
Results. The identity of the isolates should be presented at the beginning of this section (as well as the rationale for identification) and only then phenotypic and genotypic antibiotic resistance profiles.
Figure 1 should be deleted, it is not necessary.
Page 3, at the end. It is assumed that bla genes are in plasmids, whereas this is not absolutely demonstrated and in fact recognized later in the manuscript (second paragraph from discussion).
Figure 2. Delete the part “CTX-M-15” from the figure since the PCR is not specific for CTX-M-15 but for genes encoding CTX-M-1-like enzymes (CTX-M-1, CTX-M-3, CTX-M-15, CTX-M-55,…)
Page 4. What do the authors mean with “relatively new specie of the genus Escherichia…”?
Figure 3. Why is ATCC35469 repeated? What is the size of the fragment used to construct the tree? The topology of the tree shows precisely the problems in differentiating closely related Escherichia and Shigella species (something that is recognized at the end of the discussion).
Discussion. Provide a reference at the end of the first paragraph.
Table 6. Reference 45 does not include the primers used for blaESBL screening.

Author Response
We are thankful for all the concerns and corrections highlighted by the reviewer. Our responses to all comments are as follows:
Comment 1:
Strains identification. The 16S data provided to support identification is not reliable. First, the fragment analyzed is only ~700bp long which is insufficient to provide appropriate discriminatory power. Second, the comparison of 16S sequences obtained and deposited in GenBank reveals a ~90% identity between the three isolates (with several gaps or NT basis that need to be revised and corrected) (see clustal W attached). Third, according to the authors, the 16S sequences obtained showed identical homologies (e.g. ADE4 95%) with either the type strain from E. fergusonii or Shigella sp. or other Escherichia species. So, the authors are advised to provide a larger 16S sequence (~1.5Kb) from each isolate that would provide a higher resolution and support phenotypic data. These sequences should substitute those already available on GenBank.
Reply: We are aware, that for the confirmatory identification of the organism polyphasic approach is the ideal . This involves the combination of DNA-DNA hybridization, phylogenetic characterization and numerical taxonomy. Due to limited funding we could not extend the identification using the DNA-DNA hybridization. However when we compared the results obtained from the phenotypic characterization and 16 rRNA ribotypying of our organisms was assigned to the E. fergusonii with similarity index of 95% and 98%, as indicated from the GenBank. We have reframed in the abstract and within the write-up to situate this. We deeply appreciate your queries.
Comment 2:
ESBL characterization. PCR performed on plasmid DNA extracted are not sufficient to support the presence of plasmid-mediated beta-lactamases. The extracts can have chromosomal DNA (that in most cases is not completely eliminated) which is a confounding factor. Again, other narrow spectrum TEM or SHV variants might be present. As such, sequencing of these products needs to be performed for this study and not in subsequent studies because it is needed to support results and conclusions. Otherwise, the authors need to recognize this limitation and modify the manuscript in accordance.
Reply: We have highlighted this limitation and modified the manuscript in accordance.
Comment 3:
Carbapenems resistance. OXA-48 carbapenemase genes were not searched by PCR. The primers used to search for OXA are directed for OXA-1 variants that include either narrow and extended-spetrum beta-lactamases. If the authors decide not to search for OXA-48 type carbapenemases, they need to consider this limitation in the manuscript.
Reply: We have considered this limitation in the manuscript.
Other comments
4. Abstract. The identity of the strains should be considered immediately before presenting their phenotypic properties.
Reply: Identity of obtained strains has been reported before the description of their phenotypic traits.
5. Results. The identity of the isolates should be presented at the beginning of this section (as well as the rationale for identification) and only then phenotypic and genotypic antibiotic resistance profiles.
Reply: Identity of obtained strains has been reported before the description of their phenotypic traits and antibiotics resistance profiles
6. Figure 1 should be deleted, it is not necessary.
Reply: Figure has been removed.
7. Page 3, at the end. It is assumed that bla genes are in plasmids, whereas this is not absolutely demonstrated and in fact recognized later in the manuscript (second paragraph from discussion).
Reply: To address this point, we substituted “plasmids DNA” with “DNA”.
8. Figure 2. Delete the part “CTX-M-15” from the figure since the PCR is not specific for CTX-M-15 but for genes encoding CTX-M-1-like enzymes (CTX-M-1, CTX-M-3, CTX-M-15, CTX-M-55,…)
Reply: CTX-M-15 has been deleted.
9. Page 4. What do the authors mean with “relatively new specie of the genus Escherichia…”?
Reply: E. fergusonii was recently grouped as a specie in the genus Escherichia.
10. Figure 3. Why is ATCC35469 repeated? What is the size of the fragment used to construct the tree? The topology of the tree shows precisely the problems in differentiating closely related Escherichia and Shigella species (something that is recognized at the end of the discussion).
Reply: The two ATCC35469 have different ascension numbers and nucleotide sequence length. 518 nucleotide sequences per isolate was used to construct the tree after alignment using Clustal W.
11. Discussion. Provide a reference at the end of the first paragraph.
Reply: Addressed approriately
12. Table 6. Reference 45 does not include the primers used for blaESBL screening.
Reply: Reference 48 (formerly 45 before manuscript revision) includes primers used for blaTEM, blaSHV and blaOXA.
Thank you.
Round 3
Reviewer 3 Report
Species identification is still a problem in this manuscript that the authors need to solve. I am sensitive to the economic constraints of the authors. If they are not able to perform 1 PCR, they need to revise sequences and the phylogenetic tree to demonstrate species identification clearly. Some suggestions to do that are provided to the authors.
Author Response
Response to Reviewer 2:
We thank you for your review. Our responses to all comments are as follows:
Comment 1:
Species identification is still a problem in this manuscript that the authors need to solve. I am sensitive to the economic constraints of the authors. If they are not able to perform 1 PCR, they need to revise sequences and the phylogenetic tree to demonstrate species identification clearly. Some suggestions to do that are provided to the authors.
Reply: The forward and reverse AB files for each isolates were reedited and realigned using bioedit. Following clustalW multiple alignment, consensus sequences of 1349bp (Escherichia fergusonni ADE4), 1390bp (Escherichia fergusonii ADE11) and 1385bp (Escherichia fergusonii ADE18) were created.
BLAST search on Genbank showed an increase in sequence similarity of approximately 99.33% (Escherichia fergusonni ADE4), 99.56% (Escherichia fergusonii ADE11) and 99.49% (Escherichia fergusonii ADE18).
A new phylogenetic tree showing distinct clades of members of the Enterobacteriaceae family was constructed to demonstrate a clear species identification.
Comment 2:
Conclusion
Reply: We have alligned the conclusion with the result and discussion.
Thank you.